# Vitamin D and Periodontitis: A Systematic Review and Meta-Analysis

**DOI:** 10.3390/nu12082177

**Published:** 2020-07-22

**Authors:** Vanessa Machado, Sofia Lobo, Luís Proença, José João Mendes, João Botelho

**Affiliations:** 1Clinical Research Unit (CRU), Centro de Investigação Interdisciplinar Egas Moniz (CiiEM), Instituto Universitário Egas Moniz (IUEM), 2829-511 Almada, Portugal; sofiacllobo@gmail.com (S.L.); jbotelho@egasmoniz.edu.pt (J.B.); 2Periodontology Department, CiiEM, IUEM, 2829-511 Almada, Portugal; jmendes@egasmoniz.edu.pt; 3Quantitative Methods for Health Research (MQIS), CiiEM, IUEM, 2829-511 Almada, Portugal; lproenca@egasmoniz.edu.pt

**Keywords:** vitamin D, vitamin D deficiency, 25(OH)D, periodontal disease, periodontitis, systematic review, meta-analysis

## Abstract

To explore the vitamin D levels of periodontitis patients in comparison with periodontally healthy ones, and to assess the influence of vitamin D supplementation as an adjunctive during nonsurgical periodontal treatment (NSPT). Five databases (Pubmed, Embase, Scholar, Web of Sciences, and Cochrane Library) were searched until May 2020. Mean difference (MD) meta-analysis with corresponding 95% confidence interval (95% CI) and sensitivity tests via meta-regression were used. We followed Strength of Recommendation Taxonomy (SORT) to appraise the strength and quality of the evidence. Sixteen articles were included, fourteen case-control and two intervention studies, all reporting 25-hydroxyvitamin D (25(OH)D) levels. Compared with the healthy controls, the circulating 25(OH)D levels were significantly lower in chronic periodontitis patients (pooled MD = −6.80, 95% CI: −10.59 to −3.02). Subgroup analysis revealed differences among 25(OH)D measurements, with liquid chromatography-mass spectrometry being the most homogeneous method (pooled MD = −2.05, 95% CI: −3.40 to −0.71). Salivary levels of 25(OH)D showed no differences between groups. Due to the low number of studies, conclusions on aggressive periodontitis and in the effect of vitamin D supplementation after NSPT were not possible to ascribe. Compared with healthy controls, 25(OH)D serum levels are significantly lower in chronic periodontitis patients, with an overall SORT A recommendation. Future studies are needed to clarify the effect of vitamin D supplementation and the biological mechanisms linking vitamin D to the periodontium.

## 1. Introduction

Vitamin D is a fat-soluble hormone primarily obtained from exposure to sunlight, and additionally from the diet and nutritional supplements [1,2,3]. Vitamin D is a universal term employed to describe the compound that exhibits the biological activity of cholecalciferol in animals (vitamin D3). This vitamin is a key factor to the calcium-phosphate homeostasis regulation and mineral bone metabolism [4,5]. In this sense, vitamin D increases the intestinal absorption of calcium and decreases the secretion of parathyroid hormone, which consequently decreases systemic bone resorption [6,7]. In addition, vitamin D stimulates osteoblastic bone production and alkaline phosphatase activity, optimizes bone remodeling and covers bone mass by increasing bone matrix proteins [2,3,8,9].

Public awareness about vitamin D has increased over the last years due to the prevalence of its deficiency [3,10,11,12,13,14]. Accordingly, vitamin D deficiency may play an important biological and metabolic role in reducing bone mineral density, total mineral content, and, consequently, may represent a risk factor against bone healing [3,14].

Periodontitis (PD) is a complex polymicrobial disease induced by an unbalanced interaction between the oral microbial and the individual inflammatory response [15,16]. The onset of this pandemic non-communicable disease [17,18] is characterized by gum inflammation (gingivitis), and the progression results in loss of the supporting tissues of the teeth, and, if untreated, ultimately leads to tooth loss [19,20]. Furthermore, to prevent disease progression, to minimize symptoms and possibly to restore lost tissues, it requires a combination of periodontal therapeutic modalities according to patient periodontal status. The treatment can include oral hygiene instruction, subgingival instrumentation to remove plaque and calculus, local and/or systemic pharmacotherapy and periodontal surgery [21].

The nutritional consequences of vitamin D levels on periodontal health represent a matter of interest [22,23,24,25,26,27]. Over decades, lower vitamin D levels have been associated with higher periodontal destruction and severe periodontitis stages [28,29,30,31,32,33,34,35,36]. Others supported the idea that patients with higher levels of vitamin D were related to less bleeding on probing (BoP) comparing to patients with lower levels [34]. In addition, in vitro studies demonstrated that vitamin D may decrease the number of *Porphyromonas gingivalis* through active autophagy [37] and might decrease the inflammatory burden of periodontitis in rodent models [38,39,40,41]. Furthermore, the association between vitamin D levels and periodontitis has been systematically investigated [42,43,44,45]; however, none of these studies were able to produce quantitative synthesis. Additionally, the impact of vitamin D supplementation during nonsurgical periodontal therapy (NSPT) has never been appraised in a systematic way.

Therefore, the aim of this systematic review was two-fold. The primary objective was to render robust synthesis on the association between vitamin D levels and periodontitis. The secondary objective was to assess the influence of vitamin D supplementation as an adjunctive during NSPT.

## 2. Materials and Methods

### 2.1. Protocol and Registration

This systematic review protocol was defined a priori and no deviations from the protocol were made. The work followed the Cochrane Handbook of Systematic Reviews of Interventions [46] and the report was made according to the Preferred Reporting Items for Systematic Reviews and Meta-Analyses (PRISMA) guidelines [47] (Appendix A).

### 2.2. Focused Questions and Eligibility Criteria

The following focused questions were addressed:“Are 25-hydroxyvitamin D (25[OH]D) levels associated with periodontitis?”

Chronic and aggressive periodontitis patients (Patients—P); periodontitis (Exposure—E); periodontal healthy patients (Comparison—C); serum and salivary 25-hydroxyvitamin D (25(OH)D) levels (Outcome—O).
2.“Does vitamin D supplementation have an adjunctive effect on NSPT?”

Patients with periodontitis supplemented with vitamin D (Patients—P); NSPT (Exposure—E); patients with periodontitis supplemented with placebo (Comparison—C); periodontal probing depth (PPD), clinical attachment loss (CAL), BoP levels (Outcome—O).

Inclusion criteria were determined as follows: Design: Intervention trials (randomized clinical trials (RCTs) and non-randomized studies of interventions (NRSI)) and observational studies (case-control, cohort studies);Studies in humans reporting 25(OH)D levels in patients with and without periodontitis;Studies in humans reporting the effect of vitamin D supplementation as an adjunctive of NSPT;Studies describing vitamin D units of measurement and measurement methodology.

The search was conducted without any restrictions regarding year of publication or language.

### 2.3. Search Strategy

We searched Pubmed, MEDLINE (Medical Literature Analysis and Retrieval System Online), CENTRAL (The Cochrane Central Register of Controlled Trials), EMBASE (The Excerpta Medica Database), Web of Science from the earliest data available until May 2020. We merged keywords and the Medical Subject Headings (MeSHs) regarding the periodontal disease (periodontitis OR gingivitis OR periodontal health OR periodontal diseases [MeSH]) and Vitamin D (vitamin D OR (vitamin D [MeSH]) OR 25-hydroxy-vitamin OR calcitriol OR vitamin D supplementation OR Vitamin D deficiency OR vitamin D receptor) in accordance with the thesaurus of each database. Grey literature was searched through the OpenGrey portal [48]. Additional appropriate literature was included after a manual search of the reference lists of the final included articles. Periodontology-and nutrition-specific journals were hand-searched to identify additional articles.

### 2.4. Study Selection and Data Extraction

Two researchers (S.L. and V.M.) independently selected the relevant articles through titles and abstracts and excluded unrelated studies. A third author (J.B.) checked the eligible studies and any disagreement was resolved through discussion. If there were multiple publications for the same study, data from the largest sample were used.

Two researchers (S.L. and V.M.) independently extracted the relevant data from the studies. Any disagreement was resolved through discussion with a third researcher (J.B). A predefined table was used to extract necessary data from each eligible study, including the first author’s name, publication year, the country where the study was conducted, exclusion criteria, number of participants, gender, mean age, percentage of smokers, periodontal case definition, sample type (saliva or serum), measurement of 25(OH)D levels and laboratory analysis. Clinical periodontal measures included PPD, CAL and BoP. All data were independently extracted by two reviewers with a consensus in all aspects. The authors were contacted for additional data clarification when necessary.

### 2.5. Risk of Bias (RoB) in Individual Studies

Methodological quality assessment was independently performed by two calibrated authors (V.M. and J.B.) using the Cochrane risk-of-bias tool 2 (RoB2) for RCTs [46] or ROBINS-I tool for NRSI [49]. For case-control and cohort studies, we used the Newcastle-Ottawa Scale (NOS). Regarding this last tool, we scored across three categories: studies with 7–9 stars were deemed of low RoB, studies with 5–6 stars of moderate RoB, whilst studies with less than 5 stars were deemed of high RoB. Any doubt was resolved by discussion with a third author. 

### 2.6. Statistical Analysis

Statistical analysis was performed using R version 4.0.0 (R Studio Team 2018). For continuous data, mean values and standard deviations (SD) were used and analyzed with mean differences (MD) and correspondent estimates by 95% confidence interval (95% CI). The unit of measurement used in the MD meta-analysis was ng/mL. In the case of median and interquartile range report, we converted to mean and SD following Hozo et al. [50]. DerSimonian-Laird random-effects meta-analysis [51] and forest plots were performed using the ‘meta’ package [52]. Statistical heterogeneity was inspected through the I^2^ index and Cochrane’s Q statistic (*p* < 0.1). The overall homogeneity was calculated through the χ2 test [53]. All tests were two-tailed, with alpha set at 0.05. Further, the weight percentage given to each study in each analysis was provided in the forest plots. Meta-regression was performed towards the influence of smoking in serum 25(OH)D levels. Publication bias was planned if at least 10 or more studies were included [53]. In the case of lacking data amenable to meta-analysis, we followed the Synthesis Without Meta-analysis (SWiM) guidelines to synthesize quantitative data [54].

### 2.7. Strength of Recommendations

We employed the Strength of Recommendation Taxonomy (SORT) to appraise the strength and quality of the evidence [55]. The outcomes of the present systematic review, clinical recommendations, and future necessary research were discussed.

## 3. Results

### 3.1. Study Selection

Overall, the search yielded a total of 1531 studies (Figure 1). After duplicate removal, 1460 were screened for titles and abstracts, and 125 articles fulfilled the inclusion criteria (1335 were excluded). These 125 articles were subjected to full paper review eligibility and 109 were excluded as they did not approach the research questions (Appendix A). Then, 16 articles were included for the qualitative analysis, of which two concerned Vitamin D supplementation. A total of 13 studies were included in the quantitative synthesis regarding 25(OH)D levels in patients with and without periodontitis (Figure 1).

### 3.2. Studies Characteristics

Overall, the included case-control studies were from ten different countries, across Asia, Europe and America (Table 1), with a total of 10,597 participants included in this review. Two articles [28,56] reported 91 aggressive periodontitis cases—one assessed the circulating 25(OH)D levels [28], and the other assessed salivary levels [56]. Quantitative analysis included a total of 10,506 subjects (subcategorised as 9718 periodontal healthy patients and 788 patients with chronic periodontitis) from 13 studies, of which three [33,57,58] assessed 25(OH)D levels through salivary samples, nine through serum levels [28,29,30,31,32,36,59,60,61], and one assessed both methods [62].

In Vitamin D supplementation on NSPT, two studies were included [63,64]. Overall, 557 patients suffering from chronic periodontitis were submitted to NSPT and 276 participants were medicated with Vitamin D3 supplements. Gao et al. [63] searched the effect of two different concentrations of Vitamin D3 supplements in a controlled design (Table 2).

### 3.3. Risk of Bias within Studies

Thirteen articles presented with low RoB (four with 9/9, four with 8/9 and five with 7/9 scores) (Figure 2, Appendix A). Only one article presented moderate risk of bias, with an overall score of 6/9 [32]. The main reason for bias arose from the representativeness of the cases. Overall, articles adopted an adequate periodontitis case definition (100%, *n* = 14) and definition of control. A considerable number of studies failed to include representative samples (57.1%, *n* = 8) and the selection of controls (42.9%, *n* = 6). In the ascertainment of exposure, usability of the same method of ascertainment for cases and controls, and non-response rate, all articles presented low RoB (100.0%, *n* = 14). Two intervention trials presented low risk of bias, one RCT (Appendix A) and one NRSI (Appendix A).

### 3.4. Synthesis of Results

#### 3.4.1. Vitamin D Levels and Periodontitis

##### Serum Levels

In our analysis, chronic periodontitis was associated with average lower serum 25(OH)D levels (MD of −6.80, 95% CI: −10.59; −3.02), but with high heterogeneity (I^2^ = 97%) (Figure 2). To mitigate the level of heterogeneity, we performed a subgroup analysis according to the method of 25(OH)D measurement (Figure 3). The liquid chromatography–mass spectrometry (LC-MS) method presented the lower differences and a moderate heterogeneity (MD of −2.05, 95% CI: −3.40; −0.71, I^2^ = 41%). The enzyme-linked immunosorbent Assay (ELISA) (MD of −13.69, 95% CI: −24.86; −2.53, I^2^ = 97%) and chemiluminescence immunoassay (CLIA) (MD of −4.25, 95% CI: −6.69; −1.82, I^2^ = 54%) presented higher differences but also higher heterogeneity. Univariate meta-regression found that the decrease in serum levels of 25(OH)D was not associated with smokers (Appendix A).

Due to the existence of only two articles [28,56] regarding the comparison of serum levels of 25(OH)D of patients diagnosed with aggressive periodontitis compared to healthy periodontal patients, the meta-analysis was not deemed possible (Table 1). While one study [28] reported lower levels of 25(OH)D serum levels, the other, based on salivary measurements, showed opposite results [56]. 

##### Salivary Levels

Regarding salivary 25(OH)D levels, our analysis did not report differences comparing chronic periodontitis to healthy periodontal subjects (MD of 0.45, 95% CI: −1.05; 1.96) (Figure 4). 

#### 3.4.2. Vitamin D Supplementation as an Adjunctive in NSPT

Due to the lack of data amenable to perform a meta-analysis about vitamin D supplementation on NSPT, we therefore decided to synthesize evidence without analysis. Two studies fulfilled the inclusion criteria, one RCT (Gao et al., 2020) and one NRSI (Perayil et al., 2015), both of low RoB (Appendix A).

Gao et al. [63] applied vitamin D supplementation on 360 patients with moderate or severe periodontitis following NSPT. Patients were randomly assigned to 2000 international units (IU)/d vitamin D3, 1000 IU/d vitamin D3, or placebo. The effect of vitamin D supplementation tended to be modest with limited periodontal clinical relevance and long-term efficacy towards PPD and CAL (Gao et al., 2020). 

Furthermore, Perayil et al. [64] investigated, in 82 moderate chronic periodontitis patients, if vitamin D supplementation plus calcium (Shelcal-D 500 mg calcium + 250 IU vitamin D once daily) compared to placebo following NSPT. Despite PPD and CAL having no differences between groups, the authors reported significantly better results for the vitamin D group in relation to gingival inflammation and bone density (measured using panoramic x-ray).

### 3.5. Additional Analyses

We confirmed that no publication bias was assessed in meta-analysis regarding the serum 25(OH)D levels (*p* = 0.1174) (Figure 5).

### 3.6. Reporting on Strength of Recommendation

Using the SORT recommendation, we concluded that chronic periodontitis is strongly associated with lower serum levels of 25(OH)D (SORT A) [55]. 

## 4. Discussion

### 4.1. Summary of Main Findings and Quality of the Evidence

This systematic review supported an association between serum vitamin D levels (measured in ng/mL of 25(OH)D) and chronic periodontitis, with an overall SORT A recommendation. Within the lack of the available studies, salivary levels of 25(OH)D did not present an association with chronic periodontitis. In addition, there was a scarcity of studies regarding the association of aggressive periodontitis and 25(OH)D levels, and the influence of vitamin D supplementation precluded any definitive conclusion. Analyzing the impact of smoking in serum 25(OH)D level changes in chronic periodontitis, meta-regression analysis revealed that smoking had no meaningful impact. 

Overall, the results of this systematic review support a link between 25(OH)D serum levels and chronic periodontitis. That is, patients diagnosed with chronic periodontitis presented lower serum levels of 25(OH)D than periodontal healthy patients. These results are clinically relevant considering the crucial role of vitamin D in bone maintenance and in the immune system [2,3,8,65,66]. In addition, several studies have unveiled the potential harmful impact of vitamin D deficiency on the periodontium, especially after periodontal surgery where this baseline decrease might result in undesirable outcomes [67,68].

A possible mechanism for this association whereby vitamin D reduces the risk of periodontitis is through the induction of cathelicidin [69,70,71,72,73]. The vitamin D pathway has been shown to exist in human gingival fibroblasts and periodontal ligaments cells, playing an important role in immune defense in periodontal soft tissues via the activation of the human cationic antimicrobial protein cathelicidin [69,70,71,72]. Recently, serum 25(OH)D deficiency was associated with decreased hBD−2 and cathelicidin levels in periodontal tissues in gingivitis and chronic periodontitis [73].

Several limitations, however, should be reported. Firstly, the level of heterogeneity observed was high and can limit the validity and robustness of these quantitative analyses. The lack of consistency in the periodontitis case definition precluded more robust analyses of the extent and severity of periodontitis with 25(OH)D serum levels. Thus, future studies should accommodate the up-to-date consensus [74], because of its upgraded characteristics [75,76], as well providing more in-depth data on the extent and severity of periodontitis and its relevant periodontal clinical measures (such as PPD, CAL and BoP). Secondly, the main analysis in this systematic review was derived from observational studies, that only inform the association between periodontitis and 25(OH)D serum changes. Therefore, studies with longer follow-ups are mandatory to clarify this matter. Furthermore, the included studies showed multiple quantification method of 25(OH)D levels, and this may have contributed to the heterogeneity. In the future, studies should harmonize the measurement method of 25(OH)D levels. On the other hand, there are strengths in this evidence-based study. This review was designed a priori and followed a strict protocol, updated international reporting guidelines, and had an extensive literature search.

Considering the existing evidence, this is the first review to render a magnitude effect on the association between serum 25(OH)D levels and chronic periodontitis. Overall, four systematic reviews have analyzed such association [42,43,44,77], but without success in pursuing meta-analysis. Further, these reviews have reported contradictory conclusions, wherein Van der Putten et al. (2009) found no evidence of an association of vitamin D with periodontal disease in non-institutionalized elderly people, Pinto et al. [44] and Perić et al. [43] found insufficient data to provide a conclusion, while Varela-López et al. [77] reported a potential association.

With regard to periodontal treatment, our narrative synthesis provides a small view on the potential characteristics of vitamin D supplementation on NSPT. One the one hand, the shortage of literature was already highlighted, which precludes any conclusion on the effect of serum vitamin D levels on periodontal treatment [43,78]. On the other hand, a previous review found that baseline vitamin D deficiency at the time of the periodontal treatment, especially in surgical procedures, negatively affected treatment outcomes [45]. However, more randomized clinical trials are warranted to provide a robust conclusion.

### 4.2. Clinical and Research Implications

Public awareness of vitamin D levels is high due to the worrisome prevalence of vitamin D deficiency worldwide [3,10,11]. Nevertheless, it is important to highlight that to confirm the decreased total levels of 25(OH)D in a more complete way, studies may combine total 25(OH)D with parathyroid hormone (PTH), calcium and phosphate levels [79]. Thereafter, the results of this systematic review are clinically relevant because they link these low vitamin D levels to chronic periodontitis, an inflammatory condition that figures as one of the most prevalent disease in the world [18,80]. In other words, periodontitis patients are very likely to have lower serum levels of 25(OH)D, though the clinical consequences are still unclear in their entirety, and for this reason larger and well-designed RCTs are warranted.

Vitamin D levels have been consistently associated with several systemic diseases, such as rheumatic [81,82,83], cardiovascular [84,85,86,87], diabetes mellitus [87,88,89], inflammatory bowel disease [90,91], or female-related conditions [92,93,94,95]. Further, other factors such as seasonal variation [96], race and vitamin D binding protein [97] can also affect 25(OH)D levels. Likewise, periodontitis is also a relevant risk factor in these pathologies [98,99,100,101,102,103,104,105], and a potential mediation effect of periodontitis-vitamin D may be considered and further developed in clinical studies.

Our results can also serve as clinical and research guidelines for expected differences in vitamin D levels in periodontitis patients, as well as clarifying the importance of knowing the method to measure vitamin D, as this may impact vitamin D differences. At this stage, the LC-MS measurement method showed moderate heterogeneity (I2 = 41%), regardless of the multiple periodontal diagnostic criteria observed in this subgroup; therefore, we might expect more homogeneous results when studies employ the same case definition [28,30,32]. In view of these results, LC-MS might be seen as the most consistent method to measure 25(OH)D serum levels, both in clinical practice as well as in research studies, and is in agreement with a previous reliability study on vitamin D quantification [106].

While 25(OH)D serum levels of periodontitis patients were decreased compared to healthy controls, the 25(OH)D levels in whole saliva had no significant differences, and the results of the included studies are quite heterogeneous in saliva. A reasonable explanation for such differences can be the expression of vitamin-D binding protein (DBP) locally versus systemically. In this sense, a previous study on healthy periodontal patients showed that the levels of DBP in gingival crevicular fluid were higher than serum levels [107], and therefore periodontal tissue might be another source of DBP [108]. Furthermore, the levels of DBP were found to be lower in periodontitis cases, presumably due to the lack of effective production or an increase in local consumption [107]. The results on salivary levels show some heterogeneity among the studies, in particular [33,58], as well a low number of participants, which limits the validity of this finding. A possible explanation for the discrepancy between Miricescu et al., 2014 [58] and Costantini et al., 2020 [33] is the difference between mean age, gender distribution and periodontal diagnosis between these two studies.

Notwithstanding these issues, more studies are warranted to explore the role of DBP expression on the salivary and serum levels of 25(OH)D on periodontal patients.

Mindful of the inflammation surrounding teeth, peri-implantitis is a pathological condition characterized by the progressive loss of supporting peri-implant bone [109], and strong evidence has suggested that PD is a risk factor for implant loss [110]. A single study found that 25(OH)D levels are significantly decreased in peri-implantitis patients comparing with peri-implant healthy patients and, consequently, might be important indicators of peri-implant diseases [111]. Nevertheless, further studies are needed to confirm if such an association with vitamin D follows the same fashion as PD.

Importantly, future studies on the impact of vitamin D supplementation should bear in mind the baseline status of the patients. Currently, vitamin D supplementation studies support the arbitrary application of vitamin D supplements as an adjunct in NSPT, though studies have seemed to disregard the baseline vitamin D levels, and this might lead to inevitable bias of analysis. In this sense, future studies should define a priori which patients have vitamin D deficiency (<20 ng/mL) or insufficiency (<30 ng/mL) to clarify whether the restoration of DV levels will result in superior periodontal clinical results. Moreover, studies must consider the initial periodontal status and the interplay of 25(OH)D with key periodontal clinical measures. Therefore, intervention studies using vitamin D supplementation should define with a clear threshold alike patients according to the baseline 25(OH)D levels and the periodontal status.

## 5. Conclusions

Periodontitis is associated with lower 25(OH)D serum levels. The effect of vitamin D supplementation as an adjunct of nonsurgical periodontal treatment remains unclear due to the shortage of available studies. Future studies are needed regarding the effect of vitamin D supplementation and the biological mechanisms linking vitamin D to the periodontium.

## Figures and Tables

**Figure 1 nutrients-12-02177-f001:**
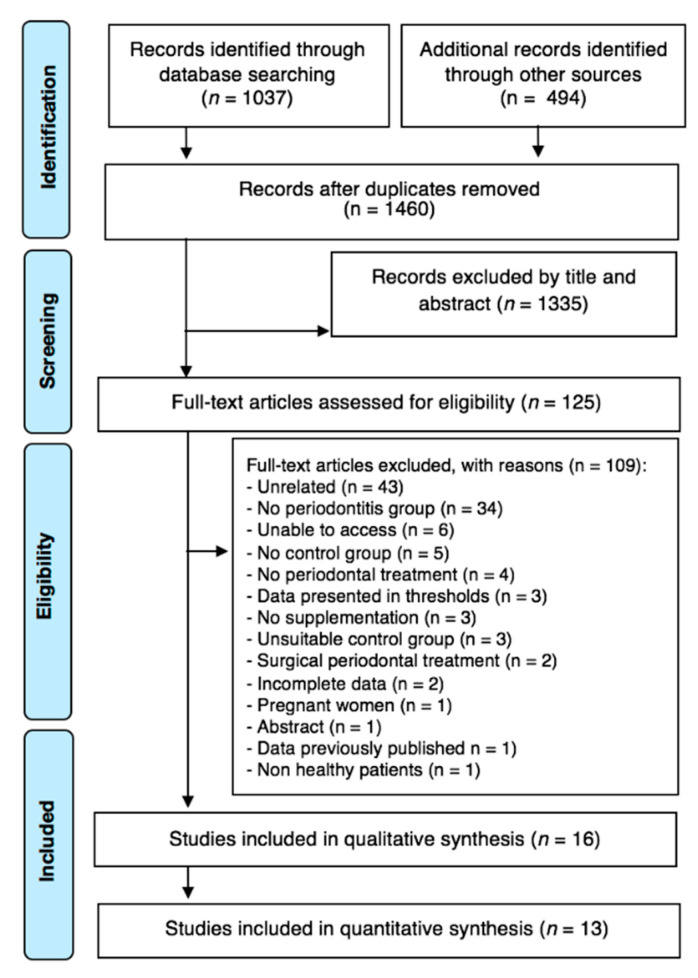
Article selection flow chart for the systematic review.

**Figure 2 nutrients-12-02177-f002:**
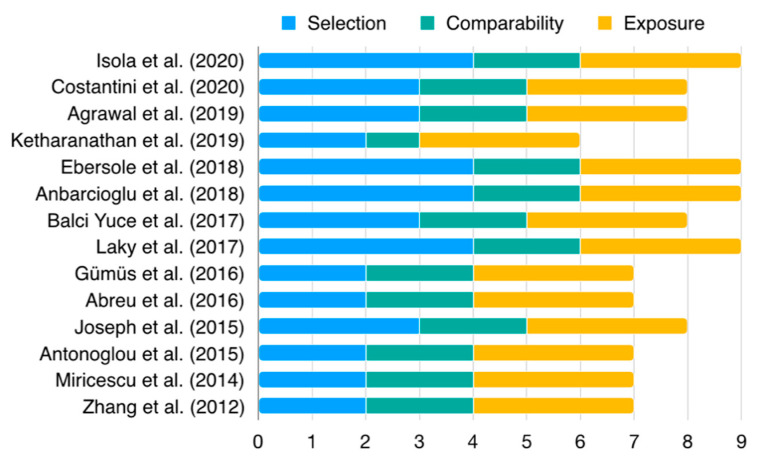
Newcastle Ottawa-Scale (NOS) for case-control studies. Detailed information is presented in the Appendix A.

**Figure 3 nutrients-12-02177-f003:**
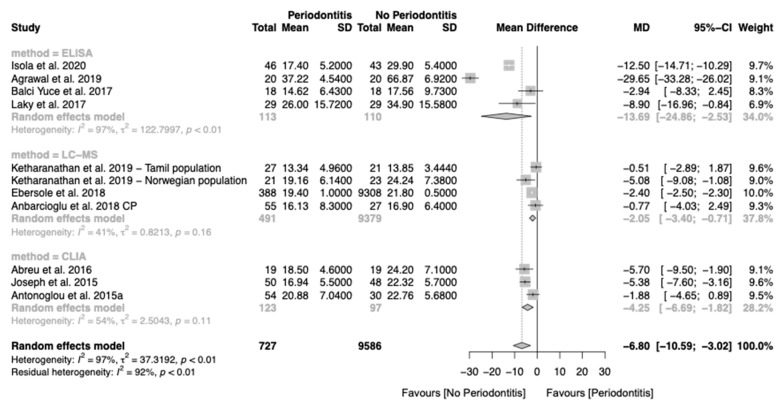
Forest plot of studies evaluating serum 25(OH)D levels in patients with and without chronic periodontitis (*p*-value < 0.001). Mean effect size estimates have been calculated with the correspondent 95% confidence intervals (95% CI). Area of squares represents sample size, continuous horizontal lines and diamonds width represents 95% CI. The diamond and the vertical dotted line represent the overall pooled estimate.

**Figure 4 nutrients-12-02177-f004:**
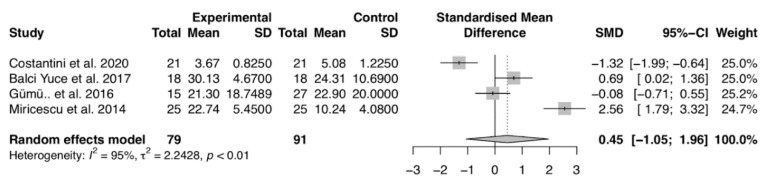
Forest plot of studies evaluating salivary 25(OH)D levels in patients with and without periodontitis (*p*-value = 0.5545). Mean effect size estimates have been calculated with 95% confidence intervals and are shown in the figure. Area of squares represents sample size, continuous horizontal lines and diamonds width represents 95% confidence interval. The diamond and the vertical dotted line represent the overall pooled estimate.

**Figure 5 nutrients-12-02177-f005:**
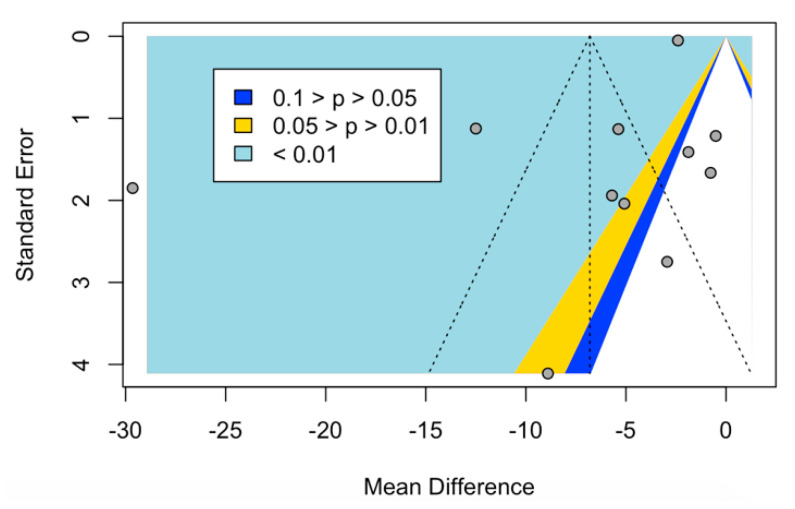
Funnel plot of studies evaluating serum vitamin D levels in patients with and without chronic periodontitis. Overall, this analysis showed no signs of publication bias.

**Table 1 nutrients-12-02177-t001:** Characteristics of the included studies regarding 25-hydroxyvitamin D (25(OH)D) levels.

Authors, Year, Country	Funding	N. of Subjects	N. of Healthy/ CP/AgP	Male/Female	Smokers *n* (%)	Mean Age ± SD	PD Diagnostic Criteria	Method	Samples
Constantini et al., 2020, Italy	None	42	21/21/-	14/28	5 (11.9)	54.3 ± 5.0/56.9 ± 5.4/-	EFP/AAP 2018	ELISA	Saliva
Isola et al., 2020, Italy	University of Catania	89	43/46/-	33/56	24 (27.9)	53.7 ± 4.5/53.1 ± 4.2/-	EFP/AAP 2018	ELISA	Serum
Agrawal et al., 2019, India	NR	40	20/20/-	NR	0 (0)	44.7/39.3/-	GI ≥ 1, PI ≥ 1, PPD ≥ 5 mm and CAL ≥ 5 mm	ELISA	Serum
Ketharanathan et al., 2019 (Tamil)	Institute of Clinical Odontology, Faculty of Dentistry, University of Oslo, Norway	48	21/27/-	48/0	2 (4.6)	41.1 ± 5.7/42.6 ± 6.7/-	CAL ≥ 6 mm in 2 or more teeth and 1 or more sites with PPD ≥ 5 mm	LC-MS	Serum
Ketharanathan et al., 2019 (Norwegian), Norway		44	23/21/-	44/0	4 (8.3)	50.3 ± 13.1/52.1 ± 9.0/-			
Ebersole et al., 2018, USA	U.S.P.H.S. grant GM103538, GM103440, and Center for Oral Health Research (University of Kentucky College of Dentistry)	9696	9308/388/-	NR	NR	NR	NHANES 1999–2004: CAL ≥ 3 mm and PPD ≥ 4 mm; 1999–2000 and 2003–2004: PPD ≥ 3 mm and CAL ≥ 4 mm; NHANES 2001–2004: Page and Eke 2007	LC-MS	Serum
Anbarcioglu et al., 2018, Turkey	Ondokuz Mayıs University Scientific Research Projects Foundation	156	27/55/74	74/82	0 (0)	30.9 ± 3.8/39.4 ± 4.7/29.9 ± 5.2	AAP 1999	LC-MS	Serum
Yuce et al., 2017, Turkey	Gaziosmanpasa University Unit of Scientific Research Projects	36	18/18/-	18/18	4 (22.2)	48.8 ± 9.6/49.5 ± 9.4 /-	AAP 1999	ELISA	Saliva and Serum
Laky et al., 2017, Austria	Austrian National Bank (*n*°. 12986)	58	29/29/-	20/38	19 (32.8)	35.5 ± 7.4/35.4 ± 7.7/-	PPD ≥ 5 mm	ELISA	Serum
Abreu et al., 2016, Puerto Rico	National Institute on Minority Health and Health Disparities of the National Institutes of Health	38	19/19/-	10/28	0 (0)	46.7 ± 8.2/47.6 ± 8.7/-	CDC—AAP 2003	CLIA	Serum
Gümüş et al., 2016, Turkey	Research foundation of Ege University, Izmir, Turkey (n°. 2013DIS029)	42	27/15/-	0/42	0 (0)	25.0 ± 4.0/-/40.0 ±10. 0	Armitage 1999	ELISA	Saliva
Joseph et al., 2015, India	SBMR, Directorate of Medical Education, Government of Kerala	98	48/50/-	46/52	9 (9,2)	40.77 ± 5.1/40.76 ± 7.8 /-	Armitage 1999	CLIA	Serum
Antonoglou et al., 2015, Finland	CIMO, Finnish Ministry of Education and Culture, Finnish Dental Society Apollonia	84	30/54/-	32/52	40 (47,6)	41.9 ± 12.7/46.3 ± 13.7 /-	Page and Eke 2007	CLIA	Serum
Miricescu et al., 2014, Romania	European Social Fund and Romanian Government (POSDRU/6/1.5/S/S17)	50	25/25/-	16/34	0 (0)	18.66 ± 2/-/51.26 ± 7.4	At least 6 sites with PPD ≥ 4 mm; bone loss > 30% and gingival inflammation	ELISA	Saliva
Zhang et al., 2012, China	Natural Science Foundations of China; National Key Project of Scientific and Technical Supporting Programs of China; Clinical Research Fund, Ministry of Health	76	32/-/44	29/47	7 (10,1)	24.3 ± 0.8 /-/ 26.8 ± 1.7	Classification of Periodontal Diseases and Conditions 1999	ELISA	Saliva

AAP—American Academy of Periodontology; AgP—aggressive periodontitis; CAL—clinical attachment loss; CDC—Centers for Disease Control; CIMO—Centre for International Mobility; CLIA—chemiluminescence immunoassay; CP—chronic periodontitis; EFP/AAP—European Federation of Periodontology/American Academy of Periodontology; ELISA—enzyme-linked immunosorbent assay; GI—gingival index; LC-MS—liquid chromatography-atmospheric pressure chemical ionization-mass spectrometry; NHANES—National Health and Nutrition Examination Survey; NR—not reported; PI—periodontal index; PD—periodontitis; PPD—probing periodontal depth; SBMR—State Board of Medical Research; SD—standard deviation.

**Table 2 nutrients-12-02177-t002:** Characteristics of the included studies regarding the 25(OH)D supplementation after nonsurgical periodontal treatment (NSPT).

Authors, Year, Country	Funding	N. of Subjects	N. of Control/CP	Male/Female	Smokers *n* (%)	Mean Age ± SD	PD Diagnostic Criteria	PD Treatment	Method	Samples
Gao et al., 2020, China	Beijing Science and Technology Program and the National Natural Science Foundation of China	240	120/120	119/121	5 (11.9)	53.0 ± 5.2/51.0 ± 6.3	At least 6 sites with PPD > 6 mm, CAL > 4 mm, X-ray showing at least 6 sites with alveolar bone loss more than one third of the root length	Control—NSPT and placebo; Test group—NSPT and 1000 IU/day vitamin D3	ELISA	Serum (Baseline and 3 months after NSPT)
		240	120/120	116/125	5 (11.9)	53.0 ± 5.2/49.0 ± 5.4		Control—NSPT and placebo; Test group—NSPT and 2000 IU/day vitamin D3		
Perayil et al., 2015, India	None	77	41/36	NR	NR	NR	One or more teeth with chronic moderate periodontitis, CAL of 3–4 m	Control – NSPT and placebo; Test group—NSPT and 60,000–120,000 IU/day vitamin D3	ELISA	Serum (Baseline and 3 months after NSPT)

CAL—clinical attachment loss; ELISA—enzyme-linked immunosorbent assay; IU—international unit; NR—not reported; NSPT—nonsurgical periodontal treatment; PPD—probing periodontal depth; SD—standard deviation.

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
