# Peer review of "Vitamin D and Periodontitis: A Systematic Review and Meta-Analysis"

_nutrients, 2020, doi:10.3390/nu12082177_

Round 1

Reviewer 1 Report

Dr. Machado et al, analyzed 16 articles, 14 case-control and 2 intervention studies,  having reporting 25-hydroxyvitamin D (25(OH)D) levels in periodontitis patients and controlled subjects. The Meta-analysis results showed the serum total 25(OH)D levels were significantly lower in chronic periodontitis patients compared with health control subjects. They suggest that future studies are needed to clarify the effect of vitamin D supplementation and the biological mechanisms linking vitamin D to the periodontium.

Main concern:

  1. In figure 4 showed there is no significant difference in salivary 25(OH)D levels in patients with and without periodontitis. In their discussion, the authors should explain why there are difference in serum and salivary levels in their discussion? Do the salivary 25(OH)D levels reflect free 25(OH)D and serum 25OHD reflect total 25OHD?
  2. What is the % of 25OHD <20ng/ml (vitamin D deficiency) and <30ng/ml (insufficiency) in patients with and without periodontitis, any significant difference? Or you should present 25OHD levels in subjects with and without periodontitis in table 1.

Minor concern:

  1. Clinical and Research Implications: Low total 25OHD level may not equal to vitamin D deficiency as the salivary 25OHD level unchanged. Combined total 25OHD plus iPTH, calcium and phos may help to diagnosis of vitamin D deficiency (Jassil et al).
  2. Zhang, Meng et al found that salivary and serum vitamin D binding protein (DBP) difference in periodontitis patients and control subject may help your discussion.
  3. Consider the lower total 25OHD and report higher DBP levels in patient with periodontitis (Zhang, Meng), suggest that the free 25OHD levels might be much lower in patient with periodontitis.
  4. Besides assay methods, seasonal difference, smoker, race and DBP polymorphisms (Youselzadeh ) will affect total 25OHD levels, you should me in your discussion.

N Jassil, et al. Vitamin D Binding Protein and 25-Hydroxyvitamin D Levels: Emerging Clinical Applications. Endocr Pract. 2017;23: 605-61

Zhang, Meng et al, Int J Endocrinol. 2014;2014:783575

P Youselzadeh. Vitamin D Binding Protein Impact on 25-Hydroxyvitamin D Levels under Different Physiologic and Pathologic Conditions. Int J Endocrinol, vol. 2014; 2014. doi:10.1155/2014/981581

Author Response

Dear Editorial Board,

We are pleased with the opportunity to revise and resubmit our manuscript “Vitamin D and Periodontitis: A systematic review and meta-analysis” (Manuscript ID nutrients-856932.R1).

We are very grateful for the editor and reviewers’ comments, all have been considered and taken into profound consideration.

Manuscript changes are highlighted in the revised manuscript. Our point-by-point responses to all comments are detailed below. We hope the revised manuscript will better suit the Nutrients, special issue “Vitamin D on Immune Function”. We are happy to consider further revisions and we thank you for your continued interest in our research.

Dr. Machado et al, analyzed 16 articles, 14 case-control and 2 intervention studies, having reporting 25-hydroxyvitamin D (25(OH)D) levels in periodontitis patients and controlled subjects. The Meta-analysis results showed the serum total 25(OH)D levels were significantly lower in chronic periodontitis patients compared with health control subjects. They suggest that future studies are needed to clarify the effect of vitamin D supplementation and the biological mechanisms linking vitamin D to the periodontium.

Main concern:

  1. In figure 4 showed there is no significant difference in salivary 25(OH)D levels in patients with and without periodontitis. In their discussion, the authors should explain why there are differences in serum and salivary levels in their discussion? Do the salivary 25(OH)D levels reflect free 25(OH)D and serum 25OHD reflect total 25OHD?

Answer: We made efforts to explore the association between salivary and serum 25(OH)D levels in patients with and without periodontitis in the discussion section, by stating “While 25(OH)D serum levels of periodontitis patients were decreased compared to healthy controls, the 25(OH)D levels in whole saliva had no significant differences, and the results of the included studies are quite heterogeneous in saliva. A reasonable explanation to such to such differences can be the expression of vitamin-D binding protein (DBP) locally versus systemically. In this sense, a previous study on healthy periodontal patients that the levels of DBP in gingival crevicular fluid was higher than serum levels [100], and therefore periodontal tissue might be another source of DBP [101]. Further, the levels of DBP were found lower in periodontitis cases presumably due to the lack of effective production or an increase of local consumption [100]. Notwithstanding, more studies are warranted to explore the role of DBP expression on the salivary and serum levels of 25(OH)D on periodontal patients.” (Page 4-5, Lines 317-331).

  1. What is the % of 25OHD <20ng/ml (vitamin D deficiency) and <30ng/ml (insufficiency) in patients with and without periodontitis, any significant difference? Or you should present 25OHD levels in subjects with and without periodontitis in table 1.

Answer: We agree that future studies should group patients according to the vitamin D status. Nevertheless, the included studies did not specify information, fact that was discussed in more detail by stating “. In this sense, future studies should define a priori which patients have vitamin D deficiency (<20ng/ml) or insufficiency (<30ng/ml) to clarify whether the restoration of DV levels will result in superior periodontal clinical results.” (Page 5, Lines 342-343)

Further, we opted to not include the 25(OH)D levels in patients with and without periodontitis in table 1 because this information was already in Figures 3 and 4, and therefore preventing duplication of information. However, if the editor find it relevant, we are totally available to add this information in table 1. 

Minor concern:

  1. Clinical and Research Implications: Low total 25OHD level may not equal to vitamin D deficiency as the salivary 25OHD level unchanged. Combined total 25OHD plus iPTH, calcium and phos may help to diagnosis of vitamin D deficiency (Jassil et al).

Answer: We agree with the reviewer and we added this information in section 4.4. Clinical and Research implication, by stating “Nevertheless, it is important to highlight that to confirm the decreased total levels of 25(OH)D in a more complete way, studies may combine total 25(OH)D with parathyroid hormone (PTH), calcium and phosphate levels [Jassil et al. 2017]” (Page 4, Lines 295-297).

  1. Zhang, Meng et al found that salivary and serum vitamin D binding protein (DBP) difference in periodontitis patients and control subjects may help your discussion.
  2. Consider the lower total 25OHD and report higher DBP levels in patient with periodontitis (Zhang, Meng), suggest that the free 25OHD levels might be much lower in patient with periodontitis.

Answer: We acknowledge the reviewer for the suggested article and we added the information about the implication of vitamin D-binding protein on 25(OH)D levels in the discussion section, as previously mentioned in point 1. 

  1. Besides assay methods, seasonal difference, smoker, race and DBP polymorphisms (Youselzadeh) will affect total 25OHD levels, you should be in your discussion.

Answer: We agree with the reviewer and we added this information by stating “Further, other facts such as seasonal variation [88], race and vitamin D binding protein [89] can also affect 25(OH)D levels.” (Page 4, Lines 305-306).

Reviewer 2 Report

The authors performed a meta-analysis and analyzed the connections between vitamin D levels and periodontitis. The topic of the manuscript is of high interest since periodontitis is a widespread disease that can also be associated with further diseases, such as heart attack etc. In the introduction, the authors should add some information about the current standards and protocols concerning non-surgical and surgical periodontitis therapy. Since periodontitis and peri-implantitis are similar diseases, that authors should also add information whether vitamin D levels might also be associated with peri-implantitis. Next, the authors should add come detailed clinical conclusions. For example, is vitamin D level associated also with a recommendation for surgical or non-surgical periodontitis therapy? 

Author Response

Dear Editorial Board,

We are pleased with the opportunity to revise and resubmit our manuscript “Vitamin D and Periodontitis: A systematic review and meta-analysis” (Manuscript ID nutrients-856932.R1).

We are very grateful for the editor and reviewers’ comments, all have been considered and taken into profound consideration.

Manuscript changes are highlighted in the revised manuscript. Our point-by-point responses to all comments are detailed below. We hope the revised manuscript will better suit the Nutrients, special issue “Vitamin D on Immune Function”. We are happy to consider further revisions and we thank you for your continued interest in our research.

REVIEWER 2:

The authors performed a meta-analysis and analyzed the connections between vitamin D levels and periodontitis. The topic of the manuscript is of high interest since periodontitis is a widespread disease that can also be associated with further diseases, such as heart attack etc. In the introduction, the authors should add some information about the current standards and protocols concerning non-surgical and surgical periodontitis therapy. 

Answer: We agree with the reviewer on expanding the benefit of including information about periodontal treatment in the introduction section since one of the main aims is to assess 25(OH)D levels after periodontal treatment. So, we added the following: “Further, to prevent disease progression, to minimize symptoms and possible to restore loss tissues, it requires a combination of periodontal therapeutic modalities according to patient periodontal status. The treatment can include oral-hygiene instruction, subgingival instrumentation to remove plaque and calculus, local and/or systemic pharmacotherapy and periodontal surgery [21].” (Page 2, Lines 50-54). 

Since periodontitis and peri-implantitis are similar diseases, that authors should also add information whether vitamin D levels might also be associated with peri-implantitis. Next, the authors should add come detailed clinical conclusions. For example, is vitamin D level associated also with a recommendation for surgical or non-surgical periodontitis therapy? 

Answer: We appreciate this important suggestion. We added this information to the Discussion section, stating: “Mindful of the inflammation surrounding teeth, peri-implantitis is a pathological condition characterized by the progressive loss of supporting peri-implant bone [102] and strong evidence suggested that PD is a risk factor for implant loss [103]. A single study found that 25(OH)D levels are significantly decreased in peri-implantitis patients comparing with peri-implant healthy patient and, consequently, might be important indicators of peri-implant diseases [104]. Nevertheless, further studies are needed to confirm if such association with vitamin D follows the same fashion as PD.” (Page 5, Lines 332-337)

Round 2

Reviewer 1 Report

The authors addressed most of my comments except one comment.

Author Response

Dear Editorial Board,

We are pleased with the opportunity to revise and resubmit our manuscript “Vitamin D and Periodontitis: A systematic review and meta-analysis” (Manuscript ID nutrients-856932.R2).

We are very grateful for the editor and reviewers’ comments, all have been considered and taken into profound consideration.

Manuscript changes are highlighted in the revised manuscript. Our point-by-point responses to all comments are detailed below. We hope the revised manuscript will better suit the Nutrients, special issue “Vitamin D on Immune Function”. We are happy to consider further revisions and we thank you for your continued interest in our research.

REVIEWER 1:

The authors addressed most of my comments except one comment.

Answer: Regarding the reviewer comment, we believe that the reviewer is referring to the topic in point 2. We added this information to the Discussion section, by stating “A reasonable explanation to such to such differences can be the expression of vitamin-D binding protein (DBP) locally versus systemically. In this sense, a previous study on healthy periodontal patients that the levels of DBP in gingival crevicular fluid was higher than serum levels [100], and therefore periodontal tissue might be another source of DBP [101]. Further, the levels of DBP were found lower in periodontitis cases presumably due to the lack of effective production or an increase of local consumption [100]. Notwithstanding, more studies are warranted to explore the role of DBP expression on the salivary and serum levels of 25(OH)D on periodontal patients.” (Page 11-12, Lines 318-325).

We hope we could note the lacking comment and answer it appropriately. However, we appreciate the reviewer improvement and we are totally available to answer other questions that the reviewer may have.

Reviewer 2 Report

.

Author Response

The reviewer did not make any comments.